# Capacity Development for the future of Ocean Prediction

Lillian Diarra (1), Romane Zufic (1), Audrey Hasson (1), Cécile Thomas-Courcoux (1), Enrique Alvarez Fanjul (1)

[1]Mercator Ocean International, Toulouse, France

*Correspondence to*: Lillian Diarra (ldiarra@mercator-ocean.fr)

**Abstract.** Capacity development in ocean prediction refers to the process of strengthening the abilities of individuals,
institutions, and systems to generate, access, understand, and apply ocean prediction tools and information. This encompasses building human capital, enhancing technical skills, improving physical and digital infrastructure, reinforcing governance, fostering collaborative partnerships and networks, ensuring inclusive participation, and providing sustained support—both financial and human—to ensure that ocean prediction services are effective, inclusive, and sustainable, especially in developing and vulnerable regions. The first section of this paper provides an overview of key global frameworks for capacity development
in ocean science, with a particular focus on ocean prediction. It also identifies existing gaps in current efforts. In the second part of the paper, we present the capacity development plans of the OceanPrediction Decade Collaborative Centre (DCC), developed within the context of the existing global framework. These plans are informed by the results of a dedicated survey (summarised in this paper) and are further supported by the regional project - Ocean Prediction Enhancement in Regions of Africa (OPERA). This section emphasises the importance of integrating both technical and non-technical training, fostering
community building, engaging stakeholders, and undertaking complementary actions to create an enabling environment for capacity development. It also highlights the value of a co-design approach and the need for continuous evaluation of the effectiveness and long-term impact of these initiatives. Finally, the discussion section offers recommendations for the future, drawing on the work carried out under the OPERA project and aligned with capacity development guidelines from the Intergovernmental Oceanographic Commission of UNESCO and the United Nations Decade of Ocean Science for Sustainable
Development.

## 1 Introduction

Capacity development is defined by the Intergovernmental Oceanographic Commission of UNESCO (IOC-UNESCO) as "the process by which individuals and organisations obtain, strengthen, and maintain the capabilities to set and achieve their development objectives over time" (UNESCO-IOC, 2021a). The IOC Group of Experts on Capacity Development describes

the goals of capacity development as "achieving evenly distributed capacity across the globe, across generations, and genders, thus reversing asymmetry in knowledge, skills, and access to technology" (IOC-UNESCO, 2020). Capacity development is thus a polysemic notion, which shows its uncharted extent when considering that "components of capacity include knowledge, skills, systems, structures, processes, values, resources and powers that, taken together, confer a range of political, managerial and technical capabilities" (Shackeroff, Theisen et al. 2016). In the context of ocean science, capacity development is described

as a "multifaceted process aimed at building the human, institutional, technical, and financial abilities needed to conduct, understand, and apply ocean science for sustainable development" (Harden-Davies et al., 2022). Capacity development thus extends beyond knowledge dissemination and training, and encompasses the strengthening of physical and digital infrastructure, advancement of technology, improvement of data accessibility, establishment of sustainable funding mechanisms, and fostering collaborative networks and participatory decision-making. These priorities are underscored in the

United Nations Decade of Ocean Science for Sustainable Development (Ocean Decade) White Paper Challenge 9: Skills, knowledge, technology and participatory decision-making for all (Arbic et al., 2024). Such a comprehensive approach is essential to empower all stakeholders to contribute meaningfully to ocean science and governance, crucial to achieve Ocean Decade Challenge 9 and underpinning progress across all other Ocean Decade Challenges.

The Ocean Decade has made capacity development one of its main priorities, a key for delivering "the ocean we want" (IOC-UNESCO 2020). Strengthening countries' capacities in building and sustaining ocean observing systems is decisive to inform and guide policymaking, and to develop and implement international agreements for a sustainable ocean (Miloslavich et al., 2018). Ocean Decade challenge 9 thus seeks to "ensure comprehensive capacity development and equitable access to data, information, knowledge and technology across all aspects of ocean science and for all stakeholders" (UNESCO-IOC, 2021b).

The equity principle is crucial, as the Global Ocean Science Report demonstrated over the years the extent to which inequalities persist in ocean science, whether in geographical, gender, or generational representations (IOC-UNESCO, 2017; IOC-UNESCO, 2020).  Indeed, studies tend to demonstrate that capacities are continuously larger in developed regions than in developing regions, as illustrated in Figure  analysing the number of ocean science publications per country. Also, scientific cooperation across regions, despite intensifying, remains too limited within developed countries from Europe, North America, and Asia (IOC-UNESCO, 2020). Strengthening the capacities of these groups, while pursuing, to a larger extent, developing

ocean science skills and knowledge of all, is the two-fold aim of capacity development in the Ocean Decade. When it comes to gender and generational imbalances, the Global Ocean Science Report demonstrated that women and young ocean scientists continue to be underrepresented in ocean science (Black 2020; IOC-UNESCO, 2020).

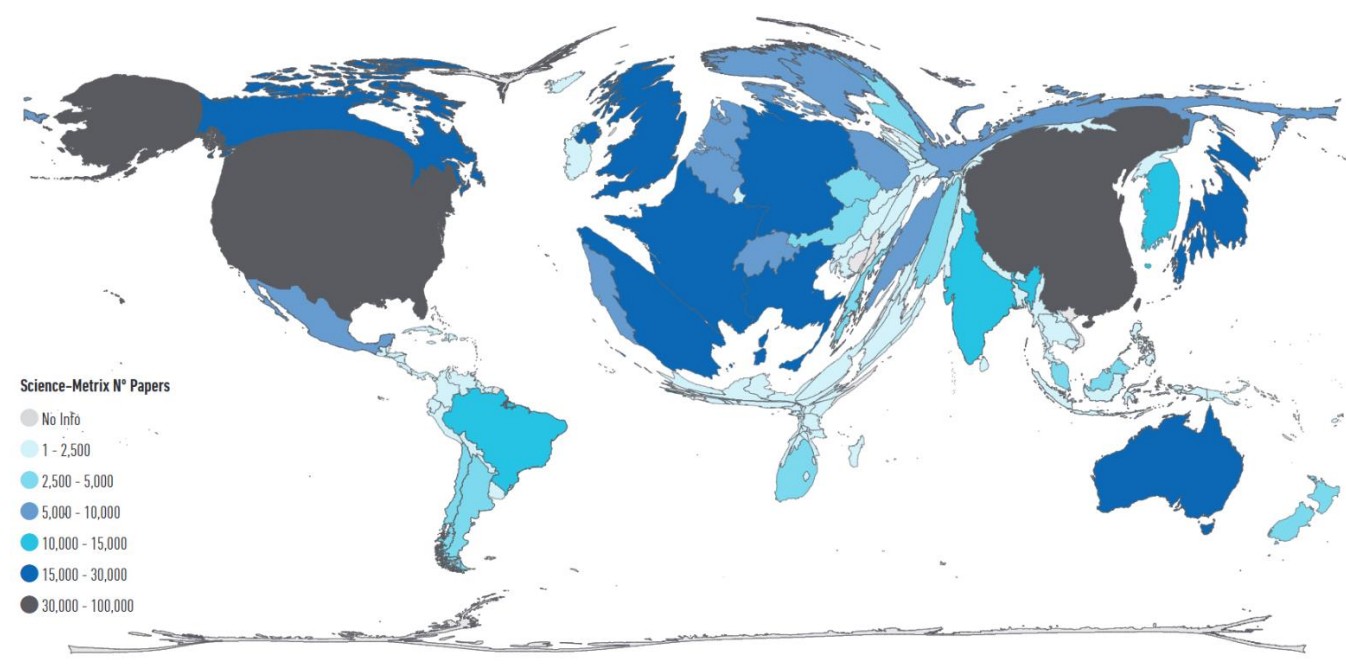

Science–Metrix N° Papers
- No Info
- 1 - 2,500
- 2,500 - 5,000
- 5,000 - 10,000
- 10,000 - 15,000
- 15,000 - 30,000
- 30,000 - 100,000


*Figure 1: Distorted world map showing each country scaled in proportion to the number of ocean science publications. Different colours indicate different numbers of publications. Source: IOC-UNESCO, 2017*

Given this situation, capacity development for ocean forecasting is more relevant than ever. This paper explores the actual
status and the plans outlined in the framework of the OceanPrediction Decade Collaborative Centre (DCC). The first section provides an overview of the current capacity development landscape, including a review of global frameworks and platforms in ocean science that are relevant to ocean prediction. The second section summarizes the findings of a survey conducted by the OceanPrediction DCC, which gathered insights on current practices and needs in the field. The analysis of these results informed the design of capacity development activities within the newly launched OPERA project – Ocean Prediction
Enhancement in Regions of Africa. This project is being implemented under the guidance of the OceanPrediction DCC's Africa Regional Team. The next section outlines a three-step approach to understanding capacity development, as framed by the OceanPrediction DCC. The discussion section presents recommendations based on the OPERA project's capacity development strategy and implementation plan. These are aligned with the guidelines of the IOC-UNESCO and Ocean Decade.

It should be noted that, while ocean literacy is an essential component of capacity development, it is beyond the scope of this paper and is therefore not addressed in this review.

## 2. Present status:  main capacity development efforts in ocean science

### 2.1.  IOC-UNESCO activities

In early 2023, IOC-UNESCO launched the Ocean CD-Hub (https://oceancd.org/, last access: 14/05/2025) to openly share worldwide ocean-related capacity development opportunities, posted by any stakeholder willing to contribute. The platform classifies the opportunities into different types, responding to the diversity of activities mentioned above. The Ocean CD-Hub also allows sorting the opportunities through regions and stakeholders. Out of 422 referenced opportunities currently, more than two-thirds are proposed by academic and research stakeholders, or by international and intergovernmental agencies. The remaining activities are proposed by governmental parties, private sector stakeholders, nonprofit and philanthropic organisations. These results may evolve as the platform continues to develop, yet it still provides a clear indicator of the main actors involved in ocean science capacity development.

IOC-UNESCO is further advancing its capacity development objectives through the Ocean Teacher Global Academy (OTGA), a flagship initiative aimed at delivering high-quality training and education in ocean science and services, implemented by the International Oceanographic Data and Information Exchange (IODE) programme and through the IOC Sub-Commissions and Regional Committees (Claudet et al., 2020; Miloslavich et al. 2018). OTGA courses have a specific focus on IOC's Member States' training needs, with special attention to developing countries (but not only) and ensuring, during the applications' selection, a gender-balanced representation in its courses, as per UNESCO's gender policies. An endorsed project of the Ocean Decade, OTGA has developed a strong international network of local universities and research institutes, acting as regional training centres (Figure 2). These centres develop courses addressing regional training needs, aligned with the IOC's policies and guidelines. Additionally, it enables training in the regionally relevant languages and resorting to in-field experts. OTGA, together with the European Copernicus Marine Service and EUMETSAT, also organises regular online courses to train future teachers (Supporting Marine-Earth Observations educators) and therefore multiply its impact over time.

The Ocean Decade Network (https://forum.oceandecade.org/, last access: 14/05/2025) is another global platform sharing numerous capacity development opportunities, as it references all Decade Actions, Contributions, Programmes and Projects; for example, the above-mentioned OTGA initiative is an endorsed action under the Ocean Decade. The platform enables the sorting of activities by Ocean Decade Challenges, with the most relevant being Challenge 9.  "Skills, knowledge and technology for all" and Challenge 7 "Expand the Global Ocean Observing System" working to "ensure a sustainable ocean observing system across all ocean basins that delivers accessible, timely, and actionable data and information to all users." The platform is further organised in thematic groups, including one on capacity development, to enable discussion and information exchange among peers.

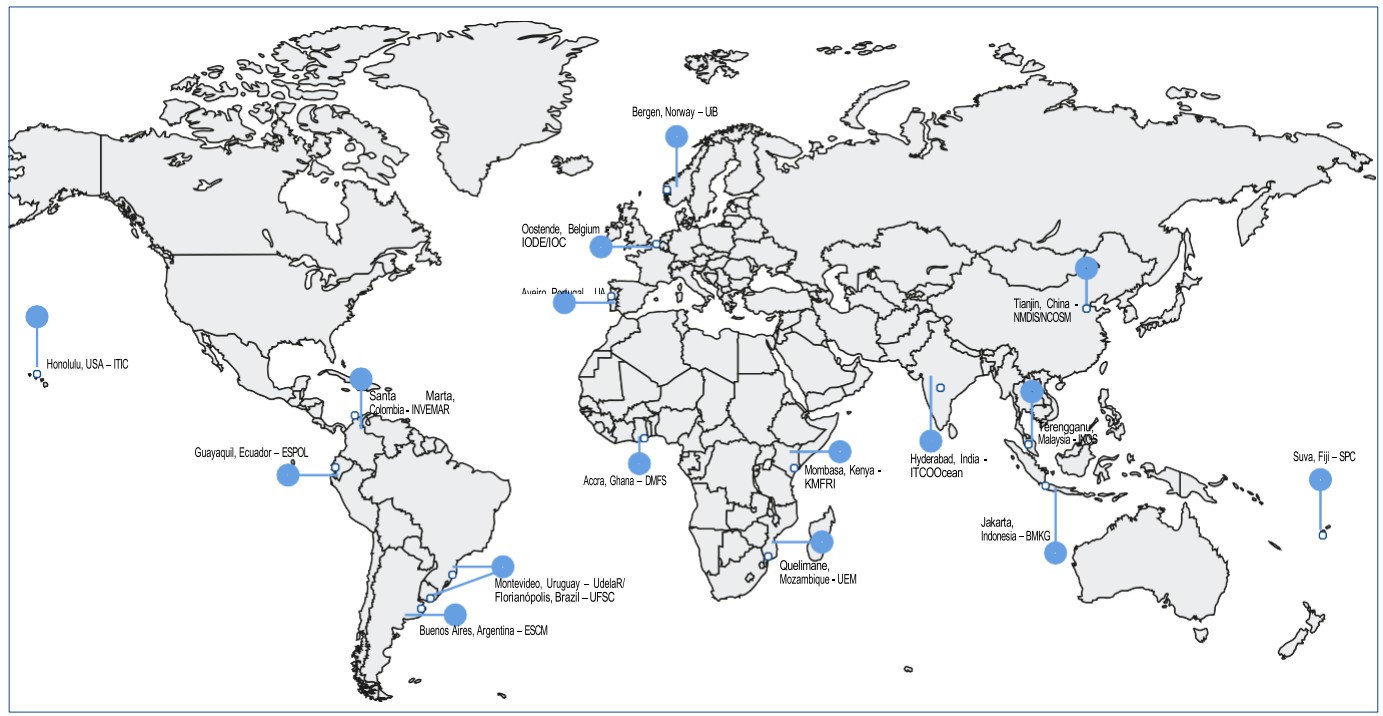

*Figure 2: OTGA regional training centres and specialized training centres in 2020. Source: IOC-UNESCO, 2020.*

The Global Ocean Observing System (GOOS) programme, which is coordinated by IOC-UNESCO, the World Meteorological Organisation (WMO), the UN Environment Programme (UNEP) and the International Science Council (ISC), comprises 15 Regional Alliances that play a key role in advancing ocean observing systems at the regional level. These alliances also lead and support targeted regional capacity development activities. Furthermore, the GOOS 2030 Strategy highlights capacity development as a top priority for strengthening in all countries, particularly those with limited resources, in order to achieve a truly integrated and inclusive global ocean observing system by 2030 (Fisher et al., 2019). In 2022, GOOS and its Expert Team on Operational Ocean Forecasting Systems (ETOOFS), with support from IOC-UNESCO, the World Meteorological Organisation (WMO), and Mercator Ocean International, published the published the ETOOFS Guide on "Implementing Operational Ocean Monitoring and Forecasting Systems. This reference guide aims to promote the global development, enhancement, and long-term sustainability of operational ocean monitoring and forecasting systems worldwide and delivers international standards and best practices (Alvarez Fanjul, et al., 2022). WMO also supports capacity development in marine meteorology and ocean services through regional marine centres and dedicated training programmes.

**2.2. Other capacity development initiatives**

Numerous other actors are proposing activities at the global level. This section presents some of the main initiatives in this line, but it does not provide an exhaustive list. Moreover, the initiatives presented target strengthening primarily focus on

skills. However, as aforementioned, capacity development extends to addressing challenges such as inadequate infrastructure, funding limitations, restricted data accessibility, and inequitable participation.


- The European Copernicus Marine Service (https://marine.copernicus.eu/, last access: 14/05/2025) regularly organises online training workshops on how to access and use its data. Each training is dedicated to a specific region, or the European sea basins or the other continents, and provides use cases on local applications. The trainings are also tailored to different themes, known as the Copernicus blue markets, demonstrating how ocean data can inform and

support decision-making across political/governance, socio-economic or environmental fields. The trainings are evaluated and improved through feedback surveys. Copernicus Marine also actively contributes to external training initiatives, collaborating with partners such as EUMETSAT, EMODnet, GMES and Africa, and the ECOP Decade Programme. Through these partnerships, Copernicus Marine brings valuable expertise and resources related to ocean monitoring and forecasting, helping to build capacity in the effective use of marine data and services. As part of its

efforts to support capacity development within the private sector, Copernicus Marine has organised and taken part in several ocean-data related hackathon events designed to foster innovation, entrepreneurship, and the practical use of marine data. Lastly, Copernicus Marine also proposes on-demand mentoring initiatives, tailored to specific audiences, from a week-long course in 2023 for the Masters Ocean, Atmosphere and Climate Sciences (Oceanography & Applications track) in Cotonou, Benin, to on-demand mentoring.

- The Early Career Ocean Professionals Programme (ECOP) of the Ocean Decade aims to support young professionals by providing them with a global network and ensuring knowledge transfer, opportunities for sharing, and collective participation in the international ocean dialogue. In 2020, the Programme launched a survey to which 1400 ECOPs replied, stating that network and information, and training and mentoring were among their top needs and expectations. Organised in Regional and National Nodes and Task Teams, the ECOP Programme intends to directly

develop but also promote relevant training events and mentoring opportunities for ECOPs worldwide.

- The International Ocean Institute (IOI) is active in ocean capacity development, and particularly ocean governance, since its creation in the 1970s; it organises (online) training courses, master programmes, summer schools, tailored workshops, offers scholarships and sponsorships, etc. The trainings mostly target developing countries and focus on regional perspectives; they are conducted at the national level through national training centres and partners, in the

respective country's main language.

- The Partnership for Observation of the Global Ocean (POGO) and the Scientific Committee on Oceanic Research (SCOR) are two international nonprofit organisations with capacity development activities regarding ocean observation, particularly towards developing countries. Founded in 1999, POGO implements various training programmes for early career scientists from developing countries, especially the 10-month operational oceanography

programme of the Nippon Foundation-POGO Centre of Excellence, dedicated each year to 10 postgraduate students, or the Visiting Fellowship programme - in partnership with the Scientific Committee on Oceanic Research (SCOR).

The latter was founded in 1957 by the International Science Council (ISC) to foster interdisciplinary research related to the ocean. Among its capacity development activities, it particularly organises the Visiting Scholars Programme, supporting ocean scientists to teach and provide mentoring in developing countries' ocean science institutions. Such partnership programme (also organised by POGO some years ago) had revealed providing several long-term benefits among which avoiding a 'brain-drain' from early career scientists, enabling the visiting scientists to gain a better understanding of the existing gaps in the hosting countries, and likely increasing their willingness to pursue their involvement (Urban and Seeyave, 2021).

- GEO Blue Planet, the ocean and coastal arm of the Group on Earth Observations (GEO) has capacity development as one of its core action areas with the aim to strengthen and transfer capabilities to ensure stakeholders can effectively use ocean and coastal observational data. The initiative organises training workshops around topics covered by its working groups, including ocean monitoring and prediction to support fisheries, coastal hazards, Sargassum inundations, among others. It also collaborates with stakeholders to co-design and co-develop adapted tools and services to meet specific information needs, such as the Sargassum Information Hub providing information on Sargassum monitoring and forecasting at the global, regional and national levels.

- Universities and academic institutions are not addressed in this study since countless of them around the world organise some capacity development activities as part of their higher education programmes; but they are, evidently, key players in training future professionals of ocean science or ocean governance (Miloslavich et al. 2018). Also, numerous capacity development activities exist at the local and regional levels, and it takes a strong knowledge of the regional organisation and its main stakeholders to thoroughly identify these structures and initiatives, similar to the analysis of Marine Studies Programmes in the Pacific Islands conducted by Veitayaki and Robin South (2001).

## 3. OceanPrediction DCC Global Survey on Capacity Development

The OceanPrediction DCC has established Capacity Development as one of the main tasks since its inception. To design a strategy for this objective, the OceanPrediction DCC launched a survey focusing on capacity development for ocean prediction. The survey served to assess awareness and knowledge of existing capacity development opportunities, better understand needs, gaps and interests, and to identify capacity development efforts around the globe.

The survey was completed by over 100 respondents, with 44% representing governmental agencies, 40% academic sector, 20% the private sector, 11% non-governmental organisations and 3% intergovernmental organisations. It is important to note that most responses came from technologically advanced countries in Europe and North America, which may bias the results toward more mature capacity development needs. Key findings from the survey analysis include:

- Limited awareness of existing resources: Overall, knowledge of current capacity development tools is low. Only 35% of respondents were aware of the ETOOFS Guide, and similarly low awareness was reported for other initiatives such as OTGA (52%) and the Ocean CD platform (30%). The most recognised initiative was the Ocean Decade network (68%). These results underscore the urgent need to raise awareness of existing tools and learning platforms.
- Learning about downstream applications: The most preferred approach is learning through success stories (59%), followed by guidance on accessing relevant data (53%), and hands-on training focused on specific applications and software (50%), such as oil spill modelling or water quality forecasting.
- Learning about building operational forecasting services: The most in-demand topics (61%) involve advanced techniques, including dynamic coupling, ensemble forecasting, and artificial intelligence. This is closely followed by interest in developing and operating full ocean forecasting service chains (60%).
- Expectations from capacity development activities: The highest priority for participants (66%) is networking (such as meeting experts, panellists, and fellow participants), followed by direct interaction with domain experts.
- Preferred duration and format of educational activities: Respondents showed a strong preference for short events (1 to 5 days, not necessarily consecutive). In terms of delivery format, 49% preferred hybrid events, 33% favoured online-only sessions, and 18% preferred in-person formats.

Based on these findings, the following recommendations for future OceanPrediction DCC capacity development activities can be drawn:

- Activities should align with the foundational objectives of the OceanPrediction DCC, using the ETOOFS Guide and system architecture as a central framework.
- Collaboration with Ocean Decade Programmes will be essential for success.
- In the short-term, raising awareness of existing resources—particularly the ETOOFS Guide and the best practices it offers—is a critical priority.
- Strategic partnerships with established platforms such as IOC's Ocean Teacher Global Academy and Ocean Best Practices System (OBPS) are recommended, leveraging complementary strengths to amplify the impact of capacity development initiatives. Support from universities is also advised to provide academic grounding for new specialised courses and graduate programmes aimed at training a new generation of professional ocean forecasters.

Building on this survey, the OceanPrediction DCC is currently launching a new set of surveys to gather insights from experts and stakeholders on the current state, challenges, and prospects of ocean forecasting services specific to different regions. The pilot regional survey was launched in April 2024, focusing on the African marine community. 134 responses were collected, with 60% coming from experts and users affiliated with African institutions (OceanPrediction Decade Collaborative Centre, 2024). Although the survey was not solely focused on capacity development, it emerged as one of the key priorities for enhancing ocean forecasting and its application in Africa – alongside (and instrumental to), community building, development

of new forecasting services, and applications and efforts to strengthen user uptake and societal engagement for long-term and meaningful impact. In the section on cross-cutting and additional needs, capacity development is the highest priority, even more so than dedicated funding for ocean forecasting and high-resolution services. In the elaboration of responses, this is linked to the need for a sustainable knowledge base and preparing a new generation of experts and scientists in ocean

forecasting to provide African solutions for African problems. When analysed by region (North (coastal countries from Morrocco to Egypt), West (Senegal to the Republic of the Congo), South (Angola to Mozambique), East (Tanzania to Eritrea)), the importance of capacity development is comparatively lower in the Southern region than in other regions; nevertheless, it remains the top priority, as indicated in Figure 3 extracted from the survey.

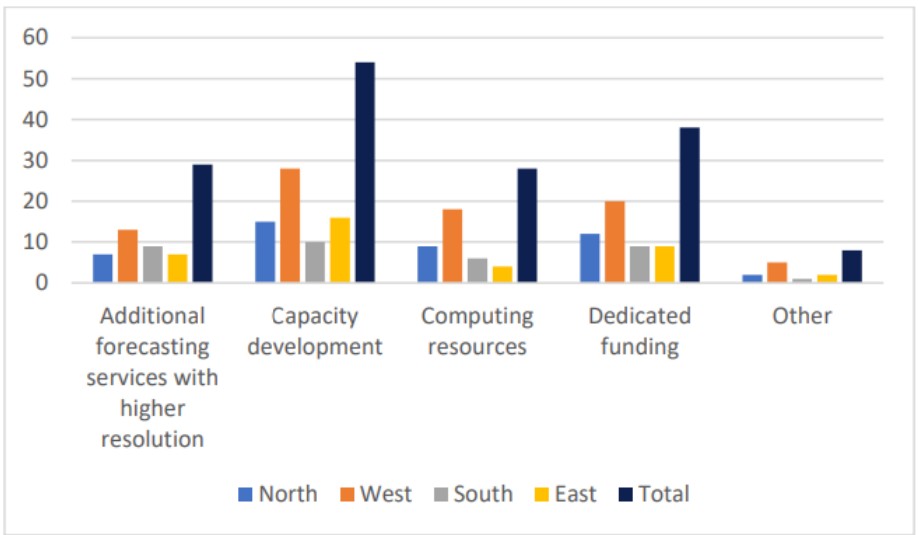

*Figure 3: Additional needs related to ocean forecasting by region (North, West, South, East) and including the total for Africa (Only replies coming from African institutions are considered). Source: Summary Results from the OceanPrediction Decade Collaborative Centre, 2024.*

In the elaboration of responses, strong emphasis was placed on engaging students and young scientists through scholarships, training to build human capital in ocean forecasting. There is a need to focus training specifically on operational oceanography,

from modelling, to data assimilation and visualisations, and should also include "training of the trainers". Respondents highlighted the importance of democratising ocean science by actively including underrepresented groups (such as women, youth, and persons with disabilities) in capacity development efforts. Additionally, the need for improved technological infrastructure and robust data management practices was recognised as a critical component of sustainable capacity development. The results from the survey helped shape the project OPERA – Ocean Prediction Enhancement for Regions in

Africa, which will be presented in the next section.

**4. Implementing capacity development activities in the OceanPrediction DCC: the OPERA project**

In January 2025, a new project was launched, called OPERA (Ocean Prediction Enhancement in Regions of Africa), within the framework of the OceanPrediction DCC and its Regional African Team. Funded by the European Union, through its ArcX programme – Support to African Regional Centres of Excellence for the Green Transition, OPERA is implemented by Mercator Ocean International through its role as coordinator of the Ocean Prediction DCC and leveraging its expertise and leadership in Copernicus Marine Service and the European Digital Twin Ocean. At its core, OPERA will strengthen ocean prediction capabilities and cooperation in Africa by supporting the development of Regional Centres of Excellence and Digital Ocean Centres, organised in three consortia, each consisting of up to five African institutional partners. These Centres will design, develop, deliver, and use fit-for-purpose ocean forecasting systems across a range of Essential Ocean Variables and build innovative ocean knowledge-based solutions to serve the needs of decision makers, coastal communities, blue economy actors, and other beneficiaries.

Following recommendations from the IOC-UNESCO framework on capacity development, the Ocean Decade Africa Roadmap, the OceanPrediction DCC ocean forecasting surveys on capacity development and African ocean forecasting survey, and consultations with various stakeholders, the OPERA capacity development strategy was co-designed to be crosscutting in the project. It encompasses community building, facilitates knowledge exchange, technological transfer to co-design innovative digital solutions, and targeted training for the consortia partners, as well as broader opportunities open to the wider African marine community. In addition, the project will support the acquisition of essential hardware for the consortia partners to strengthen their operational capabilities. This strategy will not only ensure engagement from the start but also ensure the sustainability of the action and its long-lasting impact.

The OPERA capacity development strategy is twofold. The first component focuses on capacity development activities specifically tailored to the African partners in the three consortia involved in OPERA, while the second targets the broader African marine community, with opportunities open to all interested participants. This second component aims to grow the ocean forecasting community beyond the OPERA project, essential for scaling engagement and ensuring long-term impact. A blended approach will be carried out that combines in-person and remote training, ensuring accessibility and flexibility. Together, these activities respond to Ocean Decade Challenge 9, to ensure "comprehensive capacity development and equitable access to data, information, knowledge, technology, and participatory decision-making across all aspects of ocean science and for all stakeholders" (Arbic, 2024).

## 4.1. Capacity development targeting the African marine community at large and beyond

These capacity development activities will follow the OceanPrediction DCC's virtuous loop of ocean forecasting systems and will be implemented through OPERA (Figure 4). These activities will be available online to ensure broad participation in Africa and on a global scale:

- In the first step of the loop is the Expert Team on Operational Ocean Forecast Systems (ETOOFS) Guide, which will serve as a backbone to implement activities to provide a strong theoretical foundation on ocean forecasting and its applications (Alvarez-Fanjul et al., 2022).

- The second step focuses on the Ocean Forecasting Architecture Guide and develops activities on how to build an ocean forecasting system, describing the required tools and data standards, and all the required "wiring" between the different components to ensure interoperability (Alvarez-Fanjul et al., 2024a).

- Third, the Operational Readiness Level and its associated best practices serves to develop activities train participants to operate, evaluate and improve ocean forecasting services (Alvarez-Fanjul et al., 2024b).

- Last, demonstrations via use cases and other approaches will be used to develop activities to train participants to apply ocean forecasting in real-world scenarios and integrate data into interoperable systems, with a focus on Digital Twins, particularly the European Digital Twin of the Ocean.

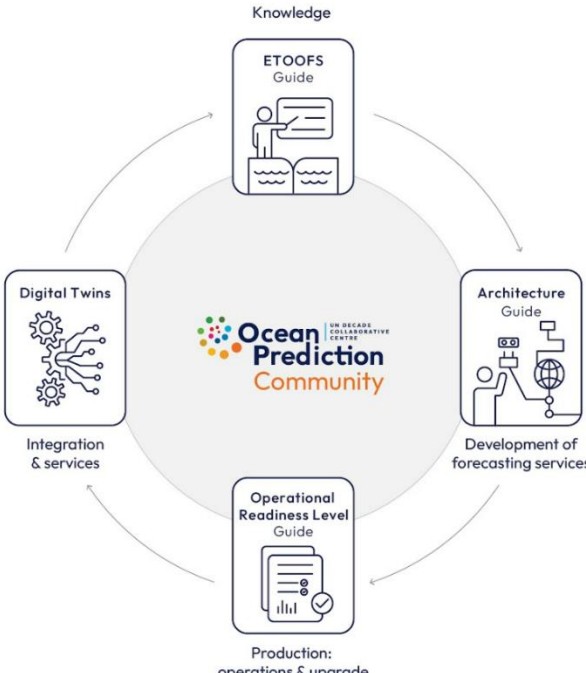

*Figure 4: OceanPrediction DCC virtuous loop for ocean forecasting (Alvarez-Fanjul et al., 2024a)*

The activities target three audiences: i) Basic Level: General Public, ii) Intermediate Level: Technical Audience with an Interest in Ocean Forecasting, and iii) Advanced Level: Experts and Practitioners Developing and Operating Ocean Forecasting Systems, adapted for multi-stakeholder participation. The intermediate and advanced levels will have associated mentoring activities, providing participants opportunities for questions and exchange. There will also be an online dedicated OPERA forum on the OceanPrediction website to facilitate discussion and knowledge exchange among project participants and also open to the African and global community at large.

The capacity development activities derived from this loop, and oriented to the described levels, are summarised as follows:

- **Ocean literacy** activities targeting non-experts to raise awareness and provide a general understanding on importance of ocean forecasting and its applications in the context of the OPERA project and the OceanPrediction DCC

- **Four Massive Online Open Courses via the OTGA**, accompanied when required by additional online lectures and introductory-level data analysis workshops, focusing on each part of the aforementioned OceanPrediction DCC's ocean forecasting virtuous loop, with increasing levels of difficulty. These courses will be available in French and English, adapted for the African context with relevant use cases. There will be a certification on completion of each MOOC.

- **Development of advanced interactive learning tool** - SEA-FORWARD (Simple Educational Access for Forecast and Warning Developers), designed to provide hands-on experience in setting up a basic ocean forecasting service. The software will serve as an educational tool, enabling users to explore forecasting methodologies, data integration, and operational workflows in a simplified yet realistic environment. There will be a certification on completion of the training.

**4.2. Capacity development specifically for OPERA project participants**

OPERA will establish three consortia of African centres led by African Institutions through competitive and open calls open to coastal countries in Sub-Saharan Africa. The first consortium will develop regional and coastal ocean forecasting systems. The other two consortia, which will be selected with attention to geographical balance, will concentrate on developing tailored applications and tools based on African priorities and regional needs. Through open calls, OPERA will establish technical expert teams to provide targeted assistance to the consortia, collaborating with them to co-design and co-develop software and digital solutions based on specific needs. The members of the technical expert team will provide tailored capacity development to the members of the consortia, so they will be able to understand, operate, and provide evolution to the ocean forecasting services and the applications developed at OPERA and implemented at Africa. The project will also organise two in-person technical training courses for consortium participants, along with annual workshops to bring together consortia members, with

325 decision-makers, users of ocean forecasting systems, and other relevant stakeholders focused on raising awareness of the project, collecting feedback and promoting the uptake of its results.

## 5. Discussion

During OPERA and after the project's implementation, it will be critical to evaluate the impact of capacity development activities—an essential cross-cutting component of the project—to assess whether they have effectively enhanced skills,
knowledge, infrastructure, ocean governance, data accessibility, and participatory decision-making at institutional, national, regional, and pan-African levels. This will be carried out through impact surveys with project participants and stakeholders involved in the project:

- Concerning the ocean literacy activities, materials will first be piloted through selected user/stakeholder groups to test comprehension and engagement levels, making necessary adjustments based on feedback. At the end of each
335 year, assessment tools, such as surveys and interactive quizzes, will be developed to evaluate the impact and effectiveness of the Ocean Literacy materials.
- Regarding the MOOCs and advanced training using the SEA-FORWARD education tool, all the interactive activities will be tested before launching with a subset of target users to refine the content, troubleshoot technical issues, and ensure the activities align with the intended skill development objectives. The project team will develop assessment
methods tailored to interactive learning, such as project-based evaluations, live demonstrations, and peer-reviewed assignments.

However, impact assessment will be time-bound, as support will not extend beyond the project's conclusion in December 2028, limiting opportunities for long-term feedback and evaluation of effectiveness of capacity development activities.

The strategy and implementation plan for capacity development activities in the OPERA project serves as a pilot project that
aims to be improved and adapted for other regions under the umbrella of the OceanPrediction DCC.

Based on guidelines from the existing literature, including the Ocean Decade White Paper on Challenge 9, several initial recommendations can be made to strengthen capacity development efforts within and beyond the OPERA project:

- **Establish mechanisms for long-term impact assessment**: Ensure that project outcomes are measured beyond the project's duration —to allow for a stronger assessment of impacts. This could include evaluating socio-economic impacts at the community level, particularly in areas such as disaster risk reduction, sustainable ocean-based economic activities, and efforts towards marine ecosystem health conservation.

- **Develop post-project capacity support structures**: Design and implement mechanisms to sustain capacity development after OPERA concludes. These may include mentoring schemes between consortium partners and technical assistance teams, long-term maintenance plans for hardware and software, and efforts to secure continued or additional funding.

- **Integrate "training of trainers"**: Embed a "train-the-trainer" approach within capacity development activities to enhance scalability and sustainability. This helps ensure knowledge transfer and skills development can continue independently within local contexts.

- **Integrating a maturity model for ocean practices**: Complement and strengthen the MOOC on Operational Readiness Level of ocean forecasting systems and its associated best practices, with a module on measuring the maturity of a practices descriptions and implementations, such as the model proposed by Mantovani et al. (2024).

- **Leverage regional networks and collaborations**: Engage with existing regional initiatives, institutions, and networks to develop more effective, locally relevant, and context-specific capacity development strategies. Collaborative approaches can help align efforts with regional priorities and amplify impact.

- **Foster interdisciplinary engagement**: Provide structured platforms that facilitate interdisciplinary collaboration. This supports the co-creation of solutions that address complex ocean challenges through integrated perspectives across natural and social sciences, technology, and policy.

## 6. Conclusion

This paper provides insights into capacity development for ocean science in the context of the Ocean Decade and more specifically, the OceanPrediction DCC. Using the OPERA project as a concrete example, the paper explores the project's proposal design, which places capacity development at its core. It highlights important elements such as co-design, early stakeholder engagement, the implementation of diverse activities targeting and adapted for multiple stakeholder groups, and continuous evaluation of these activities' effectiveness—key prerequisites for generating long-term, meaningful impact. However, the scope and depth of capacity development activities proposed by the OPERA project are constrained by limitations in funding and time. The paper thus puts forward recommendations grounded in existing literature to strengthen the capacity development approach in the context of OPERA, for future projects in the OceanPrediction DCC.

A future version of this paper could be broadened to include global initiatives on ocean literacy related to ocean prediction, incorporating a mapping of existing activities, identification of gaps, and documentation of good practices. The mapping of the global capacity development efforts can be expanded beyond training and knowledge dissemination, to encompass other important elements, such as data accessibility, infrastructure, funding, equitable participation, which are integral to comprehensive capacity development. Specifically for the African continent, the paper could be expanded to include a mapping of capacity development, including education and training opportunities, in ocean forecasting and operational oceanography at both regional and national levels. An overview of current programmes and networks aimed at enhancing prediction

capabilities in Africa would also add value. Furthermore, a more diverse group of co-authors will be invited to future works, particularly from countries with limited ocean forecasting capacity, to bring in their valuable perspective. To enhance the discussion on measuring impacts, the paper could include specific examples of capacity development activities in the ocean prediction field that have been effective in achieving intended outcomes, contrasted with those that have been less successful. This comparative approach could help identify factors that contribute to or hinder the effectiveness of capacity development activities.

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

**Competing interests**

The contact author has declared that none of the authors has any competing interests.

**Data and/or code availability**

No research data or software have been specifically newly implemented for the scope of this manuscript.

**Authors contribution**

Lillian Diarra redesigned the study and wrote the manuscript, originally initiated by Romane Zufic and Cécile Thomas-Courcoux. Audrey Hasson contributed to final editing. Enrique Alvarez Fanjul contributed to the writing and validation. All authors reviewed and edited final manuscript.

**Acknowledgements**

Claudia Delgado, Ghent University; Anthony B. Ndah, ECOP Africa; Chris P. Nwachukwu, ECOP Africa; Stefania Ciliberti, Nologin.