# Peer review of "Capacity Development for the future of Ocean Prediction"

_State of the Planet, 2024_

## Author Response (AR1)

**RC1**: 'Comment on sp-2024-11', Alexis Valauri-Orton, 05 Nov 2024  reply

This manuscript takes a very narrow approach to the term capacity development that I do not think is representative of the field. Capacity development does not just include knowledge-building activities but also building of physical infrastructure, partnerships, and financial infrastructure necessary for ocean forecasting. The manuscript includes the OECD definition of capacity development which includes these more broad and inclusive activities but misinterprets it. Capacity development does not focus on knowledge gaps. It focuses on capacity gaps. This is a clear distinction, as knowledge is quite often not the barrier preventing action on ocean forecasting. The three areas of capacity development outlined in section 3 are relevant but not a complete representation of capacity development strategies or needs and miss critical areas such as building physical infrastructure, developing low-cost tools, designing new methods to enable sustained observations, etc.

There are also research efforts underway in the ocean literacy space to suggest that knowledge transfer does not yield ocean-aware decision making as much as behavior-change and social marketing focused approaches. It would be useful for the authors to broaden their definition of ocean literacy to not just include "filling the knowledge gap" and to describe methods for more encouraging more ocean-aware decision making or world views. This is explored somewhat in section 2.2, but it would be best established in the paper's introduction when framing the concept of ocean literacy from the beginning.

The paper also lacks discussion regarding the critical importance of co-design for any capacity development program to be successful. Capacity development programs must be co-designed with the target population in order to ensure the needs of the target population are being met.

The paper would be much improved if it included a more diverse group of co-authors, especially including from regions with limited ocean forecasting capacity.

The groups and methods outlined in the paper are examples of high quality work and are useful to showcase for the community. However, they are an incomplete representation of the field.

Note also that figure 3 refers to the number of papers where the first author is from that country.

Reply

**Citation**: https://doi.org/10.5194/sp-2024-11-RC1

**AC1**: 'Reply on RC1', Romane Zufic, 07 Feb 2025  reply

We appreciate your detailed review and the opportunity to clarify the intent of this paper. Your comments help us refine our definitions and strengthen key aspects of the contribution.

First, we would like to clarify the context in which this paper was written. It is part of a **broader three-step analysis conducted under the OceanPrediction Decade Collaborative Centre (DCC), which examines ocean prediction from multiple angles and field expertise.** The **first step was dedicated to mapping the landscape of ocean prediction from all angles,** therefore this contribution provides a current overview of capacity development (CD) and ocean literacy (OL) in the context of ocean prediction. The second step will focus on **identifying gaps**, and the third step will address **recommendations and the way forward**. Because this is only step one, discussions on effectiveness, gaps, and solutions were intentionally left for future work. We

understand the need to clarify this context directly from the introduction and will make corresponding editions.

For example, we will adjust the introduction from: *"This contribution addresses the notions of 'ocean literacy' and 'capacity development' within the ocean forecasting field."*

To: *"This paper is part of a three-step process aiming at analysing ocean forecasting from multiple dimensions. The present study focuses specifically on mapping the landscape ocean prediction (step 1) within ocean literacy and capacity development perspectives, on their role in strengthening operational ocean forecasting systems. Comprehensive assessments of the existing gaps and formulation of recommendations to improve effectiveness will be addressed in the next publications."*

You further note that *"capacity development does not focus on knowledge gaps. It focuses on capacity gaps."* This is a key distinction, and we will revise our definition accordingly. We initially framed CD primarily in terms of skill-building, but we acknowledge that **physical infrastructure, financial resources, and partnerships** are just as critical. In Section 3, we will explicitly state: *"Capacity development in ocean prediction must address more than just knowledge dissemination; it requires strengthening physical infrastructure, data accessibility, funding mechanisms, and collaborative networks. These structural components are often more critical barriers to progress than scientific knowledge alone."*

On ocean literacy, you highlight that *"knowledge transfer does not yield ocean-aware decision making as much as behaviour-change and social marketing focused approaches."* This is an important point. While knowledge is foundational, OL initiatives must **drive behavioural and societal change, intergenerational engagement, and decision-making frameworks**. In **Section 2**, we will incorporate: *"Ocean literacy extends beyond knowledge dissemination; it must also enable behaviour change, shape decision-making frameworks, and foster intergenerational engagement. Social marketing approaches and experiential learning have been recognized as key drivers of ocean-aware behaviours."*

We also appreciate your emphasis on **co-design**, and we acknowledge that our original paper did not sufficiently highlight its importance. Since submitting the paper, we conducted the **OceanPrediction DCC Capacity Development Survey**, which collected insights from a diverse range of stakeholders worldwide. We will integrate key findings from this survey findings to strengthen this discussion.

Furthermore, we fully recognize the importance of **broader representation in authorship** and presented Figure 3 to actually illustrate current limitations. While we actively sought participation from colleagues from other regions, time constraints unfortunately limited broader contributions in this first paper. Future ones, addressing gaps analysis and the way forward, will involve the CD and OL experts from the **OceanPrediction DCC regional teams**, ensuring stronger global representation.

Reply

**Citation**: https://doi.org/10.5194/sp-2024-11-AC1

AC1: Lillian Diarra, 15 May 2025

In light of the comments, we have revised the paper to

- Exclude ocean literacy

- To bring in a more holistic view of capacity development aligned with the OceanDecade White Paper on Challenge 9 definition
- Highlight the importance of co-design of capacity development plans and activities
- Given the authors are all from Mercator Ocean International, we have revised and framed the paper in the OceanPrediction DCC and added sections on identifying capacity development needs in the DCC (via the Capacity Development Survey and African Ocean forecasting survey) and using a concrete example of the OPERA project present a capacity development strategy on strengthening ocean prediction capabilities and cooperation in Africa, as a DCC regional project, and how this aligns with existing guidelines (Ocean Decade) and put forward a discussion

RC2: **'Comment on sp-2024-11'**, Anonymous Referee #2, 18 Nov 2024  **reply**

- Ocean literacy and capacity development must be examined from a broader perspective. Capacity development should offer processes that help enhance our knowledge and understanding of the ocean and not just put as "offers processes to enhance abilities and skills for implementing solutions, achieving goals and supporting collaboration," as stated by the authors. Ocean literacy should not just bring awareness or "know-why" of human impacts on our ocean but support more informed decision-making, disseminating knowledge about the importance of preserving, protecting, and sustaining the marine environment." It should be viewed as a way of connecting people intergenerational, i.e., kids to adults, decision-makers to non-decision-makers, about the importance of the ocean to people and the planet and the need to protect it. In most societies, especially in the developing world, most people do not know about the ocean and so do not empathize with its abuse or why resources should be allocated for its study and management. In most developing countries, leaders prioritize needs other than the protection and conservation of the ocean and so, therefore, do not allocate budgets for its study and management. I believe that if a connection to the ocean were built among these leaders while they were younger, they would have a great sense of responsibility towards the ocean. The article needs to build on the UN Ocean Decade white papers on challenges 9 and 10. These are the most recent articles on capacity development and ocean literacy within the framework of the UN Ocean Decade, but both are completely missing in this paper. The paper will be more impactful if it makes key ocean literacy and capacity development recommendations, i.e., where are we with respect to OL and CD, and where should we be heading? The paper is only a literature review, which does not identify any gaps in the subject and how they can be bridged. It is unclear what actions the paper is recommending or putting forth. There is no new knowledge here.
- Reply
- **Citation**: https://doi.org/10.5194/sp-2024-11-RC2
-

AC2: 'Reply on RC2', Romane Zufic, 07 Feb 2025  reply

- We greatly appreciate your feedback, as they highlight key areas for refinement, particularly in how we frame this paper about **capacity development and ocean literacy in the field of ocean prediction**.
- We would like first to clarify the **scope of this paper**. This paper is part of a **three-step analysis** under the OceanPrediction Decade Collaborative Centre (DCC), which **examines ocean prediction from multiple angles and field expertise.** The **first step was dedicated to mapping the landscape of ocean prediction from all angles,** therefore this contribution provides a current overview of capacity development (CD) and ocean literacy (OL) in the context of ocean prediction. The second step will further analyse existing gaps and limitations, and the third step will provide recommendations and propose solutions, which we fully recognise and agree the importance of. We will revise the introduction to clarify this structure.
- While we acknowledge that CD is more than just training and skill-building, our intent was not to redefine CD as a whole but rather to explore its application within the operational context of **ocean prediction**. To clarify this, we will revise the introduction to further state that CD in ocean prediction is not limited to training but also access to ocean data, infrastructure development, and long-term sustainability mechanisms, aiming to build **long-term scientific capacity.**
- Your point that "ocean literacy should not just bring awareness or 'know-why' of human impacts on our ocean but support more informed decision-making" is well-taken and fully valid. We will update our OL definition in the abstract and Section 2 to reflect its role in fostering behavioural and societal change, enabling intergenerational knowledge transfer, and supporting evidence-based decision-making.
- Regarding the **UN Ocean Decade white papers (Challenges 9 & 10)**, these were not available at the time of submission, but we appreciate the suggestion to incorporate them now. We will particularly mention the need for equitable access to data, training, and funding (Challenge 9) and the importance of behavioural transformation in ocean literacy (Challenge 10) which were not emphasized enough.
- Reply
- **Citation**: https://doi.org/10.5194/sp-2024-11-AC2

AC2: Lillian Diarra, 15 May 2025

The paper's scope has been limited to exclude Ocean Literacy and recommendations from the Ocean Decade white paper 9 have now been incorporated. We have not highlighted challenge 10 given that Ocean Literacy is no longer covered in the paper.

The paper has been revised to bring in an example of developing capacity development activities in the framework of the DCC, starting with understanding needs for capacity development (through targeted surveys) to building a capacity development plan and strategy based, through

co-design and drawing from existing literature, as well as put forward recommendations on improvements.

**RC3**: 'Comment on sp-2024-11', Anonymous Referee #3, 06 Jan 2025  reply

While this manuscript provides an overview of ongoing ocean literacy and capacity development initiatives with a focus on UNESCO led actions. However, this review has a number of limitations in its scope and in the information presented. One major gap is the lack of a discussion of ocean data and the disperse landscape in the ocean data field which impacts capacity development. This is particularly crucial in less developed countries where modern data management practices may not be followed and ocean data use and availability is limited. This is particularly important in ocean science where the authors rightly point out there are more publications coming from developed regions.

Regarding the topic of ocean literacy, the abstract proposes a limited definition (while later the definition is expanded upon the abstract should provide a more precise definition of ocean literacy and highlight its relevance to not only youths but the broader public). In addition, there should be more of a discussion of the implications and recommendations of different actions. While there is a discussion on measuring impact etc., there is room to discuss which actions have been more or less effective and the reasons behind their effectiveness (or not). Currently, the paper only gives a surface level overview of ongoing initiatives but lacks the depth to encourage greater reflection on both the topics of ocean literacy and capacity development.

Reply

**Citation**: https://doi.org/10.5194/sp-2024-11-RC3

**AC3**: 'Reply on RC3', Romane Zufic, 07 Feb 2025  reply

We appreciate your insightful comments and the opportunity to clarify our approach.

This paper is part of a **three-step analysis** under the OceanPrediction Decade Collaborative Centre (DCC), which **examines ocean prediction from multiple angles and field expertise.** The **first step was dedicated to mapping the landscape of ocean prediction from all angles,** therefore this contribution provides a current overview of capacity development and ocean literacy in the context of ocean prediction. The second step will further analyse existing gaps and limitations, and the third step will provide recommendations. This explains why the paper **does not yet discuss effectiveness or solutions**—these aspects will be covered in later work. We will revise the introduction to clarify this structure.

Your review raises important points about **ocean data accessibility** as a core issue in capacity development. We agree that a major limitation in many regions is **not just a lack of training, but limited access to ocean data**. The revised paper will explicitly highlight this challenge in Section 3 explaining that fragmented and inequitable access to ocean data, lack of infrastructure and data-sharing frameworks, are key priority areas for future CD efforts.

We will refine the definition of OL to better align with its **broader societal role**, incorporating **behavioural change, intergenerational engagement, and decision-making support**.

You also suggest discussing the **effectiveness of different OL and CD approaches**. While this paper is primarily a **mapping exercise**, we will incorporate brief references to existing evaluations. For instance, since submitting the paper, we conducted the **OceanPrediction DCC Capacity Development Survey**, which collected insights from a diverse range of stakeholders worldwide. We will integrate key findings from this survey findings to strengthen this discussion.

Finally, we acknowledge your concern that the paper focuses heavily on **UNESCO-led efforts**. This was an intentional choice due to the **UN Ocean Decade's role in global OL and CD efforts, and to the paper's being proposed within the OceanPrediction DCC framework**, addressing the specific field of ocean prediction. We will clarify in the introduction that this short paper is not intended as, could not pretend to be, an **exhaustive review of all initiatives of capacity development and ocean literacy taking place worldwide**.

Reply

**Citation**: https://doi.org/10.5194/sp-2024-11-AC3

AC1: Lillian Diarra, 15 May 2025

We have revised the paper to leave out Ocean Literacy, a limitation of the paper mentioned in the conclusion.

We have added a new section on capacity development activities in the framework of the OceanPrediction DCC with a concrete example of the OPERA project, and how it follows recommendations put forward in existing literature in building a Capacity Development strategy. In the discussion we propose ways in which to improve the activities proposed as well as how to expand the scope of the paper in the future, including actions which have been effective and why, with concrete examples.